# Comparison of Machine Learning Pixel-Based Classifiers for Detecting Archaeological Ceramics

Argyro Argyrou [1,*], Athos Agapiou [1], Apostolos Papakonstantinou [1] and Dimitrios D. Alexakis [2]

1   Department of Civil Engineering and Geomatics, Faculty of Engineering and Technology, Cyprus University of Technology, Saripolou 2-8, 3036 Achilleos 1 Building, 2nd Floor, P.O. Box 50329, Limassol 3603, Cyprus; athos.agapiou@cut.ac.cy (A.A.); a.papakonstantinou@cut.ac.cy (A.P.)
2   Laboratory of Geophysics—Satellite Remote Sensing & Archaeoenvironment (GeoSat ReSeArch Lab), Institute for Mediterranean Studies, Foundation for Research and Technology—Hellas (FORTH), Nikiforou Foka 130 & Melissinou, 74100 Rethymno, Greece; dalexakis@ims.forth.gr
*   Correspondence: ac.argyrou@edu.cut.ac.cy; Tel.: +357-25002542

**Abstract:** Recent improvements in low-altitude remote sensors and image processing analysis can be utilised to support archaeological research. Over the last decade, the increased use of remote sensing sensors and their products for archaeological science and cultural heritage studies has been reported in the literature. Therefore, different spatial and spectral analysis datasets have been applied to recognise archaeological remains or map environmental changes over time. Recently, more thorough object detection approaches have been adopted by researchers for the automated detection of surface ceramics. In this study, we applied several supervised machine learning classifiers using red-green-blue (RGB) and multispectral high-resolution drone imageries over a simulated archaeological area to evaluate their performance towards semi-automatic surface ceramic detection. The overall results indicated that low-altitude remote sensing sensors and advanced image processing techniques can be innovative in archaeological research. Nevertheless, the study results also pointed out existing research limitations in the detection of surface ceramics, which affect the detection accuracy. The development of a novel, robust methodology aimed to address the "accuracy paradox" of imbalanced data samples for optimising archaeological surface ceramic detection. At the same time, this study attempted to fill a gap in the literature by blending AI methodologies for non-uniformly distributed classes. Indeed, detecting surface ceramics using RGB or multi-spectral drone imageries should be reconsidered as an 'imbalanced data distribution' problem. To address this paradox, novel approaches need to be developed.

**Keywords:** ceramic detection; archaeology; remote sensing archaeology; artificial intelligence; machine learning; imbalanced data distribution; drone data; UAV

## 1. Introduction

Archaeological remains, such as ceramics, can be either below the ground or on the surface. These remains are evidence of historic and pre-historic activities [1]. As stated by Orengo H.A. and Garcia-Molsosa A. (2019) [2], the dispersion analysis of surface remains provides researchers with information related to potential changes in land use or the destruction of sites.

The surface survey is a straightforward method for discerning settlement patterns and forms of past human behaviour in the landscape. In addition, this method can study the interactions between past populations and their natural environment and discover archaeological heritage for protection and management purposes in the rapidly developing and changing modern landscape. Nevertheless, traditional ground surface surveys have several limitations, including the following: (a) they are considered time-consuming, (b) their use requires training, (c) they are based on sampling mainly conducted using grids, (d) only the parts of the archaeological record, that are exposed to the land surface can be

detected, (e) methodological decisions may not be sufficient to reach the goals of the survey, and (f) certain areas cannot be surveyed due to their surface conditions, accessibility and other environmental conditions (lighting, weather, flora, fauna, etc.) [2]

In recent years, remote sensing science has been increasingly applied to support archaeological research [3,4]. The ever-increasing use of space-based remote sensing applications has been supported by the technological development and improvement of space-based sensors, spatial and spectral resolution, and the implementation of open access and the free distribution of satellite datasets (Landsat and Sentinel products) [5]. However, the traditional pattern recognition methods such as photo interpretation may prove inapplicable in archaeological research covering large areas or even searching an extensive archival dataset. A crucial factor determining surface research's success is the research methodology, which may need to be revised or more reliable. Consequently, it is difficult to accurately evaluate the results and their interpretation's validity, which affects whether the research objectives can be considered successful.

The development of remote sensing over the last 20 years has incentivised the exploration of new possibilities in archaeological research [6–23]. Archaeological research using remote sensing approaches has been prompted to exploit geospatial data systematically. In addition, the democratisation of low-altitude systems, with drones at relatively low costs, has been broadly implemented in archaeological research in the last decade, primarily for documentation cases [24]. Concurrently, archaeological computational approaches and advanced artificial intelligence (AI) algorithms, rather than desktop-based approaches, are increasingly applied in cloud-based systems [2]. AI is increasingly attracting widespread interest across various scientific disciplines due to its increasingly powerful predictive capabilities [1]. Therefore, archaeologists can more fully exploit the knowledge gathered from extensive archaeological data through AI [25–50]. This enables them to make informed decisions about conservation and protection procedures for archaeological elements. Moreover, AI helps determine the most suitable excavation points in a complex cultural landscape.

An evolution of the analytical tools used to support archaeological research occurred during the last decade. This evolution includes techniques like machine learning (ML) combined with geometric morphometry. Machine learning can make detecting archaeological remains more accurate without requiring explicit programming. Lately, artificial intelligence has also been used through deep learning (DL) [49], which processes these archaeological data based on artificial neural networks with representation learning. A recent study (2002) [51] indicated that most of the ML and DL algorithms used in archaeology are for object classification and detection. Nevertheless, the detection of archaeological structures using DL algorithms still needs to be improved, specifically when employing aerial/drone imaging. One could argue that we are relatively at the beginning of a new era of so-called "remote sensing archaeology" if we consider that all the changes mentioned above occurred in a relatively short period.

Overall, the findings of Agapiou et al. [26], together with the results presented by Orengo and Garcia-Molsosa [2], showed that the application of deep learning algorithms to Unmanned Aerial Vehicles (UAV) images can be a ground-breaking innovation in the field of archaeological research, supporting future archaeological field projects. Additionally, it offers a cost-effective option that provides faster results when applied under favourable conditions, mainly in cases where the research time is restricted. However, its success and accuracy are influenced by multiple factors. To improve both the survey design and the results, we can combine additional complementary procedures like observation methods, remote sensing and AI techniques. Consequently, archaeological remains will be accurately detected by combining remote sensing, machine learning, and deep learning techniques. This will lead to a better understanding of the close relationship and interaction between man and the environment. By studying the environment of the past, we can better approach the study of man and culture and their potential interactions with the landscape in the past.

Our study aimed to investigate the feasibility of developing a semi-automatic archaeological feature detection using artificial intelligence in UAV images (multi-spectral

and RGB). The research work of this study was implemented in a simulated field where low-altitude flights were carried out using UAV sensors. The simulated field was an area where no indication of archaeological remains existed. It was given the appearance of a real archaeological field, investigating synthetic elements with known properties like rocks, crops, slopes, soil, and ceramics. We used RGB and multispectral images in the developed methodology, applying artificial intelligence techniques to identify surface archaeological ceramics. The methodology initially included using supervised machine learning classifiers like Random Forest, Support Vector Machines, etc. Then, in a second step, improvement techniques for both data and classifiers were applied. Finally, various evaluation metrics were implemented to assess the classification performance and guide the classifier modelling. The initial results proved the existence of the "accuracy paradox" in the dataset, with an imbalanced class distribution between the archaeological ceramics and the field.

Furthermore, we aimed to answer research questions more efficiently in terms of time and accuracy of the process, compared to traditional archaeological fieldwork. The overall objective of this study was to evaluate whether using low-altitude and relatively low-cost remote sensing sensors can be efficient in detecting surface ceramics through artificial intelligence and image post-processing techniques. It is important to note that the method presented in this paper does not intend to replace archaeological surface surveys but rather to ensure that more time and resources can be allocated to automated or semi-automated technical procedures necessary for the survey.

## 2. Case Study

A simulation processing was implemented over a plot of approximately 90 m$^2$ in Alambra village in the Lefkosia District of the Republic of Cyprus (Figure 1). The survey was conducted in May 2022, during a good period of visibility for archaeological material, as the fields in Cyprus had recently been ploughed.

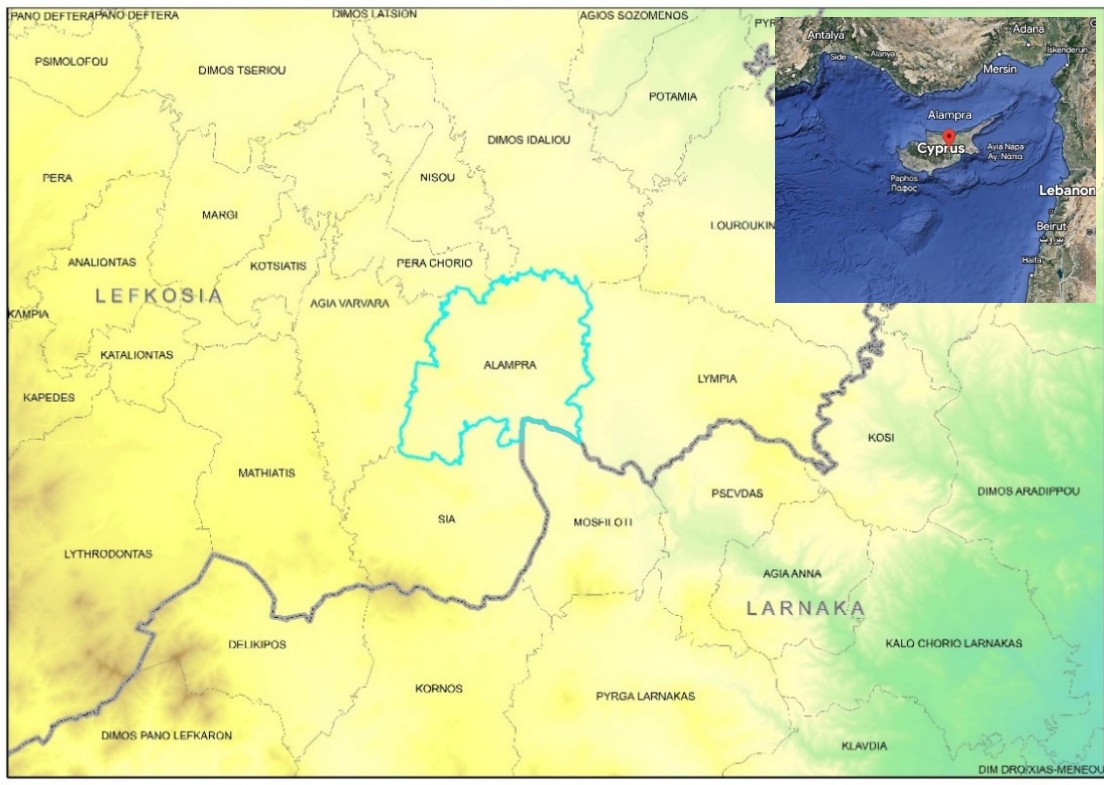

**Figure 1.** Location of Alambra village in the Lefkosia District of the Republic of Cyprus.

The field selected for the pilot study was chosen as it represented ideal field conditions during a fieldwork period in Cyprus. The area had recently been ploughed, which would

increase the visibility of ceramics, compared to fields with a extensive flora, rocks, and shadows, which can reduce the detection efficiency and cause false identifications, as soil shades resemble those of ceramics. The periodically cultivated plot corresponded to scenarios with appropriate soil visibility and offered an ideal ground for detecting ceramics (Figure 2a). This approach allowed for the evaluation of the technique's performance under the best conditions.

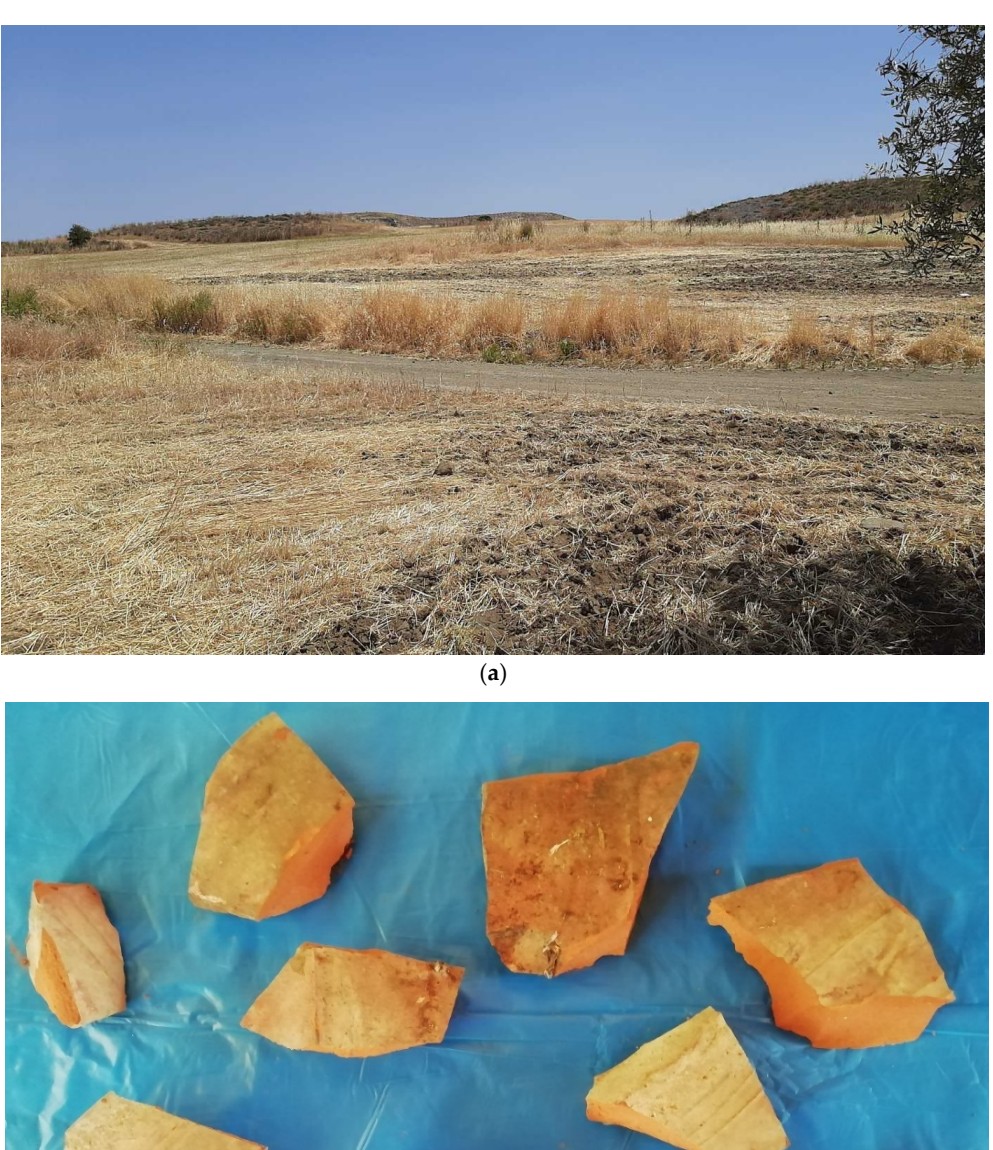

(**a**)

(**b**)

**Figure 2.** (**a**) The field selected for the pilot study (Photos: A. Argyrou, Earth Observation Cultural Heritage Research Lab©); (**b**) example of surface ceramics scattered in the field (Photos: A. Argyrou, Earth Observation Cultural Heritage Research Lab©).

The field was almost flat, with a 2% slope; no ceramic was present. For the simulation research, 365 pieces of ceramics were scattered in the field. The size of the pottery fragments ranged from 3 cm to 6 cm. The colour of the ceramics varied, from reddish-orange to brown,

depending on their firing (Figure 2b). The selected area contained no other ceramic remains than those that were placed be us explicitly for this simulation.

## 3. Materials and Methods

A combination of several recent independent technological developments was applied to the workflow upon which the research was based:

- Low-altitude and low-cost UAS have significantly improved their features and have become considerably more affordable to researchers, offering autonomy in flight time for surveying.
- Digital photogrammetry is now more user-friendly and accessible by implementing semi-automated workflows that have been integrated into many archaeological workflows [2].
- Machine learning (ML) is an element of artificial intelligence that allows software applications to be more accurate for outcome predictions, without requiring explicit programming. Machine learning applications have significantly increased in recent years and have become a usable choice for data mining, analysis, and object detection in archaeological research [1].
- Deep learning (DL), as a subset of ML learning, computers simulate human behaviour by managing data using artificial neural networks incorporating representation learning. Significant growth in this research has also occurred in recent years [1].
- Finally, various evaluation metrics were implemented to assess the performance of the classification and guide the classifier modelling.

As previously mentioned, the simulation study presented here aimed to investigate whether we could develop a semi-automatic ceramic detection methodology to answer the research questions. These questions are related to the time-consuming data processing and detection accuracy in a typical field condition. To this end, a workflow incorporated low-altitude and low-cost drone imaging for the detailed recording of the surveyed fields, as well as photogrammetry to merge all these images into one orthoimage. Finally, AI techniques like machine learning and deep learning algorithms were tested to detect and classify ceramic fragments through photomosaic. In the following paragraphs, this workflow is presented in detail (Figure 3).

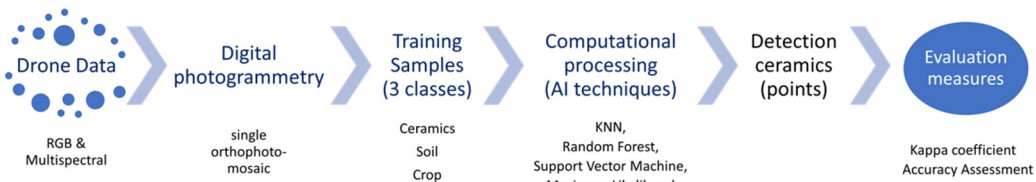

**Figure 3.** Workflow implemented in this study.

### 3.1. UAV Image Acquisition

We used two drones to acquire drone-based images of the selected area of interest. Two flight campaigns were performed on the same day using first the DJI Phantom 4 Pro system (spectral bands: Blue (B): 468 nm $\pm$ 47 nm; Green (G): 532 nm $\pm$ 58 nm; Red (R): 594 nm $\pm$ 32.5 nm), while for the second campaign, we used the DJI P4 Multispectral system(spectral bands: Blue (B): 450 nm $\pm$ 16 nm; Green (G): 560 nm $\pm$ 16 nm; Red (R): 650 nm $\pm$ 16 nm; Red edge (RE): 730 nm $\pm$ 16 nm and Near-infrared (NIR): 840 nm $\pm$ 26 nm). For both flights, the height was 20 m above ground level (AGL). The selected height provided orthophotos with a ground sample distance of approximately 2 cm/px, considered sufficient to detect ceramics on the field under survey. The flight time for each campaign was about 20 min.

*3.2. Photogrammetric Processing and Computational Processing*

The final step included computational processing (AI techniques) to identify and isolate ceramic fragments using the orthophoto mosaic of the captured images. The photogrammetric processing of the photos involved the orthorectification of all photographs and combining them into a single orthophoto mosaic using the Terra software. Orthorectifying the image involves ensuring that the images are geometrically accurate and corrected from lens distortion, camera tilt, perspective, and topographic relief. Therefore, the images were orthorectified and merged into an orthomosaic map using the photos' metadata, which contained information like drone model, types of camera sensor and lens, and GPS coordinates. After the mosaics were produced (Figure 4), image-processing techniques were applied to detect surface ceramics. The same approach was followed for both UAV flights.

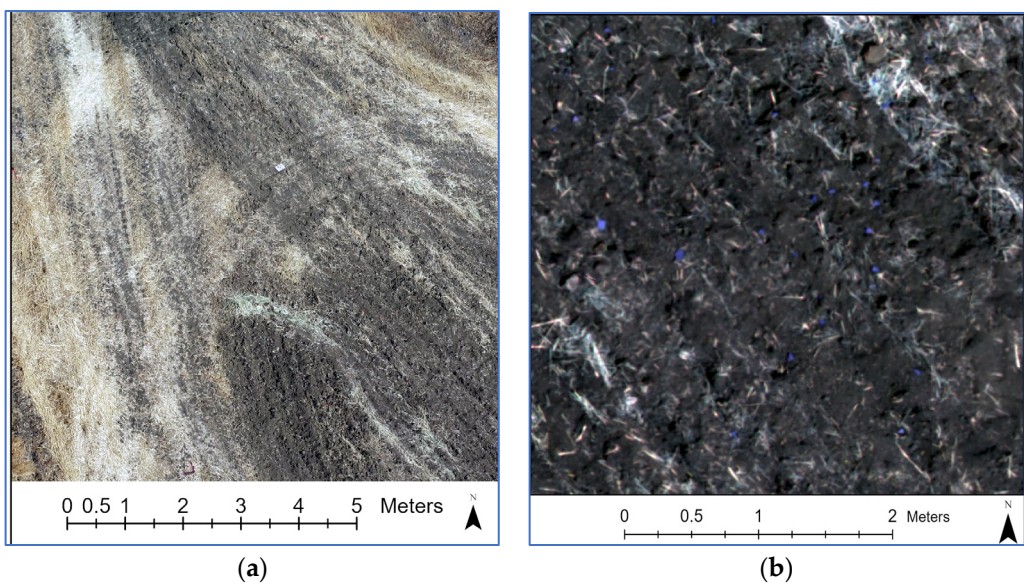

**Figure 4.** Mosaics used in this simulated study. (**a**) RGB (left) and (**b**) multispectral (Right).

ArcGIS Image Analyst tools of the ArcGIS Pro software were used for computational processing. Within the ArcGIS Pro environment, a training model was created using the Training Samples Manager in the Classification Tools, consisting of three classes: 'ceramics' (class 1), 'soil' (class 2), and 'crops' (class 3). The training sample file included a class name indicating the name of the class category and a class value containing the integer value for each class category (class 1 = 1, class 2 = 2 and class 3 = 3). The initial training data were selected by drawing polygons on top of visible ceramic fragments, bare soil, and crops. The creation of the training data consisted of assigning to each class the values of the pixels delimited by the polygons in each composite band. Four supervised classifiers (K-Nearest Neighbour (KNN), Random Forest (RF), Support Vector Machine (SVM), and the Maximum Likelihood algorithm) were applied. We set 500 samples to the SVM, RF, and KNN classifiers as the maximum number of samples per class, considering this was a high enough number to ensure optimal results. The composite images were then classified using the trained classifier. This produced the first classification output. The classification was compared to the orthomosaic to evaluate how it fitted. This step included randomly sampled points creation for post-classification accuracy assessment. The Accuracy Assessment Points tool of the Image Analyst tools was then applied to all classification results. Randomly distributed samples were created in each class, each with an equivalent number of samples. These samples were then compared with the classification results. Based on the confusion matrix per classifier, we then calculated the user's and producer's accuracy for each class, as well as the overall kappa index. This procedure

was performed for both drones' images (RGB and multispectral), while all results were extracted and evaluated on a local computer.

### 3.3. Supervised Machine Learning Classifiers

This section briefly introduces the most well-developed supervised machine-learning classifiers for detecting archaeological ceramics.

The Random Forest algorithm is a viral supervised machine learning algorithm used in many archaeological classification cases. It is based on ensemble learning and is a set of individual decision trees. Each tree combines different samples and subsets of the training data [52].

The Maximum Likelihood classifier is used for image classification. Its technique is based on two principles: the normal distribution of the pixels in each class sample in the multidimensional space and decision making using the Bayes' theorem. Assuming a normal distribution of the class sample, then each class can be indicated by a mean vector and a covariance matrix. Considering these two characteristics for each cell value, the statistical probability for each class can be assessed to define the cell's membership in the class [53].

Another supervised classifier, the Support Vector Machine (SVM), is a powerful classification method that can also process a standard image or a segmented raster input. This classification method is widely used among researchers and is trained to classify everything as the prevalent class, minimising the error and increasing the margin [54].

Finally, the K-Nearest Neighbor is another supervised classifier that can classify a pixel or segment using a plurality vote of its K neighbours. The data points in each category among these k neighbours can be counted if the Euclidean distance of the K number of neighbours is calculated [55].

The final result was compared with the number of scattered ceramics placed at the beginning of the archaeological campaign. An evaluation of the classification was also made for all classes. The results are presented in the next Section.

## 4. Results

### 4.1. Detection of Ceramics in an RGB High-Resolution Mosaic

As mentioned above, all classifiers were trained using image samples for three classes, i.e., 'ceramics' (class 1), 'soil' (class 2), and 'crops' (class 3). The overall accuracy was estimated to summarise the performance of each classification model using randomly distributed testing pixels. Accuracy is defined as the proportion of correctly predicted samples in the test set divided by the total predictions made on the test set.

**Accuracy = Correct Predictions/Total Predictions**

The accuracy for class 2 and class 3 (soil and crop) was estimated at approximately 80%, while for class 1 (ceramics), a relatively low accuracy was reported for all four classifiers. The question arising at this point was how many testing pixels should be chosen to ensure that the assessed accuracy was a reliable estimate of the actual accuracy. Would a larger sample of testing pixels give a more realistic estimate? What should the appropriate number of samples be? According to John A. Richards (2021) [56], the number of samples required for an accuracy of 90% is 225 testing pixels, while 119 testing pixels are required for a 95% accuracy. These numbers proposed by Richards assume that the classes follow a normal distribution (for each class, a set of measurements, for instance, the mean, is distributed around the centre of these measurements).

Considering the above numbers and using 225 testing samples, the accuracy of all classifiers was estimated again. ArcGIS Pro randomly created 225 sampled points for post-classification accuracy assessment using the Image Analyst Toolbox. The sampling scheme was set to randomly distributed points, in which each class had the same number of points. A "Ground Truth" field and a "Classified" field were created in the final attribute table. Finally, we manually updated the Ground Truth field by changing or identifying the

set of points, and compared these fields using the Compute Confusion Matrix tool. The results for the ceramics class varied between 12 and 24% for the RGB images, as presented in Tables 1–4.

**Table 1.** Accuracy Assessment of KNN.

| Class Value | Ceramics | Soil | Crop | Total | User Accuracy | Kappa |
|---|---|---|---|---|---|---|
| ceramics | 10 | 4 | 61 | 75 | 0.13 | |
| soil | 0 | 57 | 18 | 75 | 0.76 | |
| crop | 1 | 12 | 62 | 75 | 0.83 | |
| total | 11 | 73 | 141 | 225 | 0 | |
| producer accuracy | 0.91 | 0.78 | 0.44 | | 0.57 | |
| | | | | | | 0.36 |

**Table 2.** Accuracy Assessment of the Maximum Likelihood Classifier.

| Class Value | Ceramics | Soil | Crop | Total | User Accuracy | Kappa |
|---|---|---|---|---|---|---|
| ceramics | 9 | 12 | 54 | 75 | 0.12 | |
| soil | 0 | 57 | 18 | 75 | 0.76 | |
| crop | 0 | 8 | 67 | 75 | 0.89 | |
| total | 9 | 77 | 139 | 225 | 0 | |
| producer accuracy | 1 | 0.74 | 0.48 | | 0.59 | |
| | | | | | | 0.39 |

**Table 3.** Accuracy Assessment of Support Vector Machine Classifier (SVM).

| Class Value | Ceramics | Soil | Crop | Total | User Accuracy | Kappa |
|---|---|---|---|---|---|---|
| ceramics | 18 | 12 | 53 | 75 | 0.24 | |
| soil | 1 | 54 | 20 | 75 | 0.72 | |
| crop | 0 | 5 | 70 | 75 | 0.93 | |
| total | 19 | 63 | 143 | 225 | 0 | |
| producer accuracy | 0.95 | 0.86 | 0.49 | | 0.63 | |
| | | | | | | 0.45 |

**Table 4.** Accuracy Assessment of the Random Forest Classifier.

| Class Value | Ceramics | Soil | Crop | Total | User Accuracy | Kappa |
|---|---|---|---|---|---|---|
| ceramics | 11 | 7 | 57 | 75 | 0.15 | |
| soil | 0 | 56 | 19 | 75 | 0.75 | |
| crop | 1 | 4 | 70 | 75 | 0.93 | |
| total | 12 | 67 | 146 | 225 | 0 | |
| producer accuracy | 0.92 | 0.84 | 0.48 | | 0.61 | |
| | | | | | | 0.41 |

The distribution of the ceramics and the overall classification of all three classes (ceramics, soil, and crops) across the simulated area are depicted in Figure 5. The detected ceramics are indicated in red, while the soil is shown in green, and the crops in yellow, after implementing the supervised machine-learning classifiers referred to in Section 3.3 above.

*4.2. Detection of Ceramics in a Multispectral High-Resolution Mosaic*

Working with the multispectral dataset, the accuracy results indicated similar patterns as in the RGB datasets. For classes 2 and 3 (soil and crop), the accuracy was estimated to be approximately 90%, while for class 1 (ceramics), the accuracy was again lower. Following the same methodology as for RGB and considering the above number of 225 testing pixels, the accuracy of all classifiers was estimated. The results varied between 23% and 61% for the multispectral images (Tables 5–8), showing once more a significant decline but a better

performance compared to the classification with the RGB images. The distribution of the ceramics and the overall classification of the three classes across the simulated area are illustrated in Figure 6. The detected ceramics are indicated in red, while the soil is shown in green, and the crops in yellow.

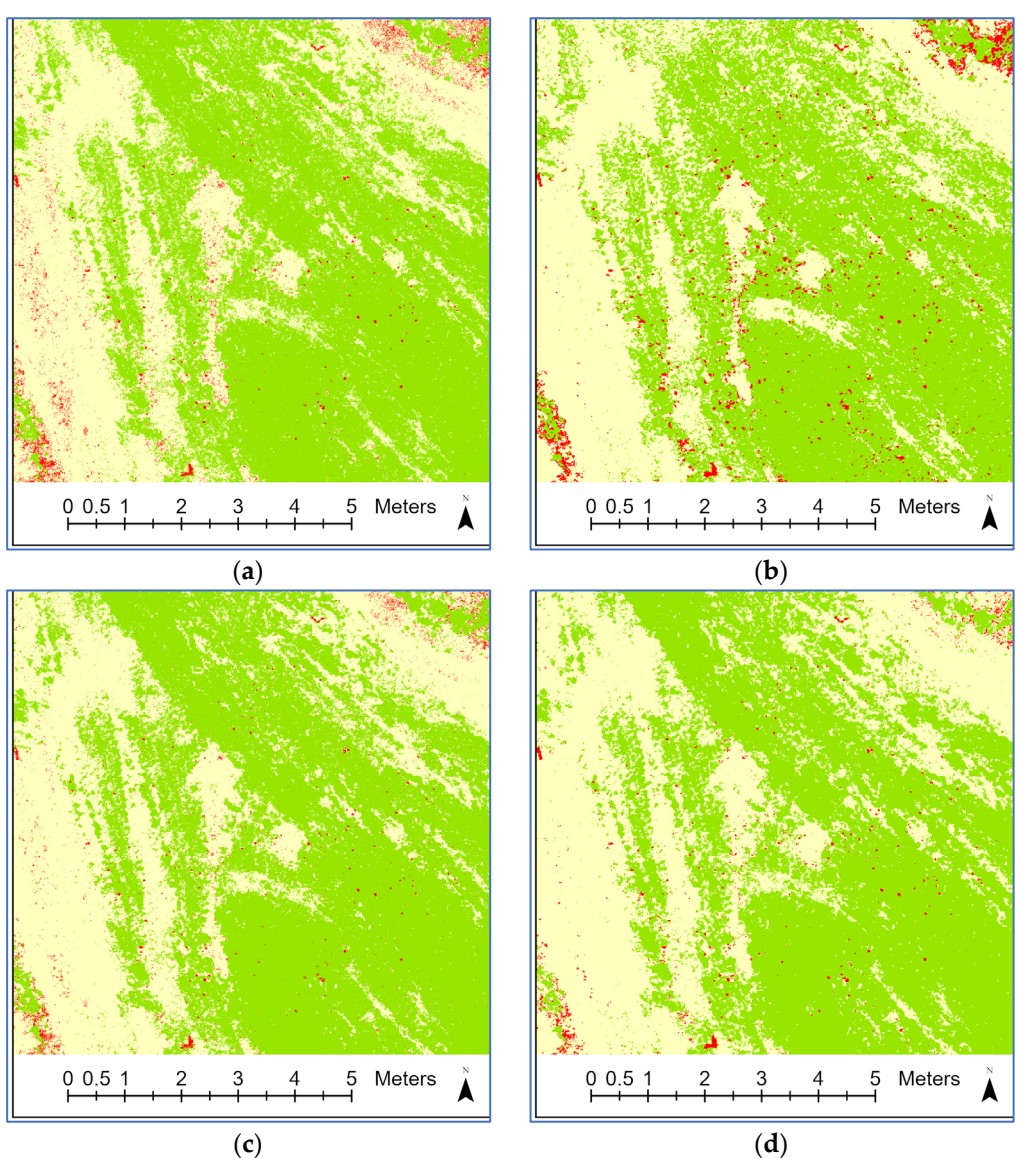

**Figure 5.** (**a**) Classification using K-Nearest Neighbor, (**b**) classification using Maximum Likelihood, (**c**) classification using Random Forest, (**d**) classification using Support Vector Machine. The detected ceramics are indicated in red, while the soil is shown in green, and the crops in yellow.

**Table 5.** Accuracy Assessment of the KNN classifier.

| Class Value | Ceramics | Soil | Crop | Total | User Accuracy | Kappa |
|---|---|---|---|---|---|---|
| ceramics | 17 | 39 | 19 | 75 | 0.23 | |
| soil | 0 | 65 | 10 | 75 | 0.87 | |
| crop | 0 | 14 | 61 | 75 | 0.81 | |
| total | 17 | 118 | 90 | 225 | 0 | |
| producer accuracy | 1 | 0.55 | 0.68 | | 0.64 | |
| | | | | | | 0.45 |

**Table 6.** Accuracy Assessment of the Maximum Likelihood Classifier.

| Class Value | Ceramics | Soil | Crop | Total | User Accuracy | Kappa |
|---|---|---|---|---|---|---|
| ceramics | 39 | 20 | 16 | 75 | 0.52 | |
| soil | 0 | 70 | 5 | 75 | 0.93 | |
| crop | 1 | 15 | 59 | 75 | 0.79 | |
| total | 40 | 105 | 80 | 225 | 0 | |
| producer accuracy | 0.975 | 0.67 | 0.74 | | 0.75 | |
| | | | | | | 0.62 |

**Table 7.** Accuracy Assessment of the Support Vector Machine Classifier (SVM).

| Class Value | Ceramics | Soil | Crop | Total | User Accuracy | Kappa |
|---|---|---|---|---|---|---|
| ceramics | 46 | 12 | 17 | 75 | 0.61 | |
| soil | 0 | 62 | 13 | 75 | 0.83 | |
| crop | 0 | 7 | 68 | 75 | 0.93 | |
| total | 46 | 81 | 98 | 225 | 0 | |
| producer accuracy | 1 | 0.77 | 0.49 | | 0.78 | |
| | | | | | | 0.67 |

**Table 8.** Accuracy Assessment of Random Forest Classifier.

| Class Value | Ceramics | Soil | Crop | Total | User Accuracy | Kappa |
|---|---|---|---|---|---|---|
| ceramics | 23 | 23 | 29 | 75 | 0.31 | |
| soil | 0 | 65 | 10 | 75 | 0.87 | |
| crop | 1 | 6 | 68 | 75 | 0.93 | |
| total | 24 | 94 | 107 | 225 | 0 | |
| producer accuracy | 0.96 | 0.69 | 0.64 | | 0.69 | |
| | | | | | | 0.54 |

A final vector point layer was then exported and incorporated into ArcGIS Pro software for visualisation and further analysis. This layer provided the number of ceramics detected automatically through the supervised classification procedure. Table 9 summarises these results (detection of the ceramics) per type of camera sensor (RGB and multispectral), underlining the spectral confusion concerning the ceramics and the ground (soil and crops). This is a common phenomenon in archaeological research. The results showed a significant divergence in ceramic detection, and the number of detected elements was not near the actual number of 365 pieces, except for the Maximum Likelihood and Support Vector Machine classifiers using multispectral images. This was also directly related to the highest number of false positive ceramic detections in all cases.

**Table 9.** Detection of ceramics with supervised machine learning algorithms using RGB and multispectral images.

| Class Ceramics | Ceramic Detection Using RGB | Ceramic Detection Using Multispectral |
|---|---|---|
| KNN | 845 | 1573 |
| Max Likelihood | 1276 | 286 |
| SVM | 794 | 250 |
| Random Forest | 548 | 705 |

An interesting observation emerges when comparing the results of all the accuracy assessments (Tables 1–8). The automated detection method that detected the higher number of ceramic fragments was the Support Vector Machine classifier, with a detection rate of 24% and a Kappa coefficient of 45% using RGB images. Additionally, 61% of the ceramic fragments were detected, with a Kappa coefficient of 67%, when using multispectral images and a Support Vector Machine classifier (Figure 7).

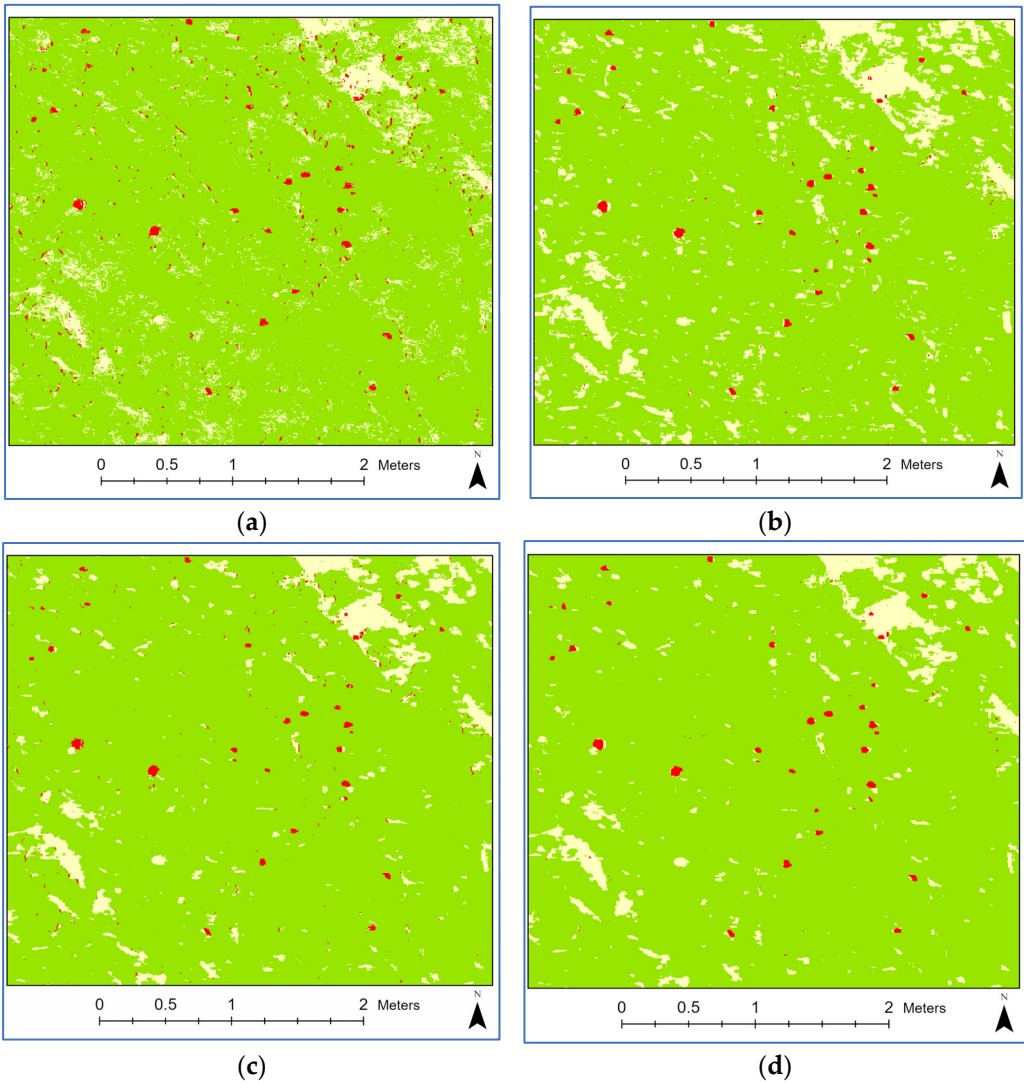

**Figure 6.** (**a**) Classification using K-Nearest Neighbor, (**b**) classification using Maximum Likelihood, (**c**) classification using Random Forest, (**d**) classification using Support Vector Machine. The detected ceramics are indicated in red, while the soil is shown in green, and the crops in yellow.

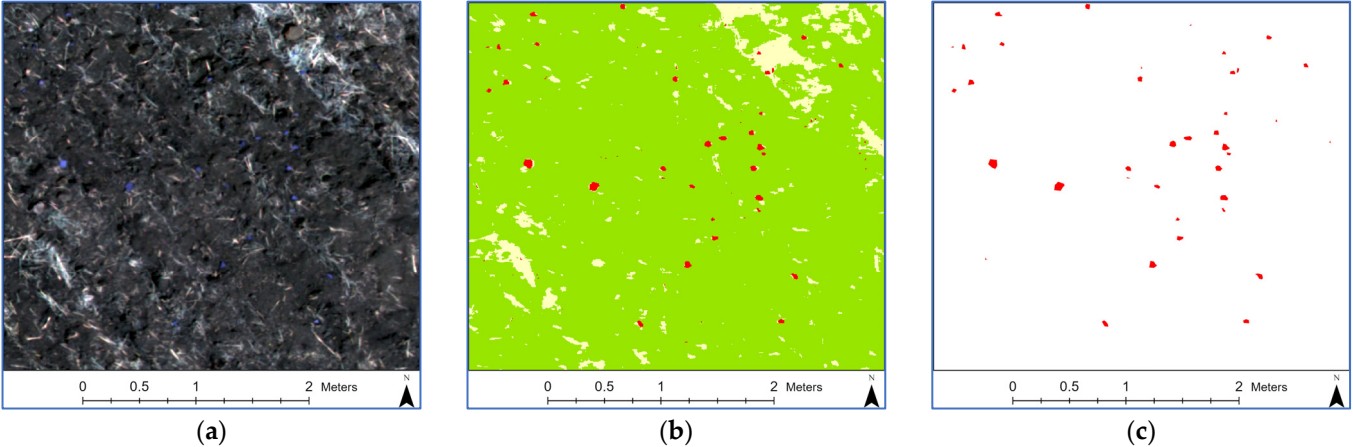

**Figure 7.** Enlarged image of the research field area. (**a**) Input multispectral image, (**b**) classification using the Support Vector Machine, with the ceramics shown in red, (**c**) detected ceramics.

## 5. Discussion

Previous results indicated that low-altitude sensors can provide significant detection results but also point out existing research limitations for detecting surface ceramics. These limitations restrict the accuracy of the detection of the minority class of ceramics. To overcome this 'accuracy paradox', future studies need to (re)consider ceramic surface detection as an 'imbalanced data distribution' problem.

Indeed, in previous studies, a problem with the misclassification of minority classes (i.e., archaeological ceramics) was found. Therefore, despite the high accuracy level, the actual detection rate for the ceramic class remained low. Classifiers tend to predict with higher accuracy classes with extensive data compared to those with few data.

Most classifiers assume a relatively balanced normal class distribution and equal misclassification costs. But when these classifiers are used to classify data with an imbalanced class distribution (skewed class proportions), their performance encounters significant drawbacks (Figure 8). In these datasets, classes with a large proportion of the dataset are called majority classes. In contrast, those with a smaller proportion are minority classes. Sun et al. indicated in 2009 [57] that the modelling can be influenced by factors besides skewed data, like a small sample size, separability, and sub-concepts within a class.

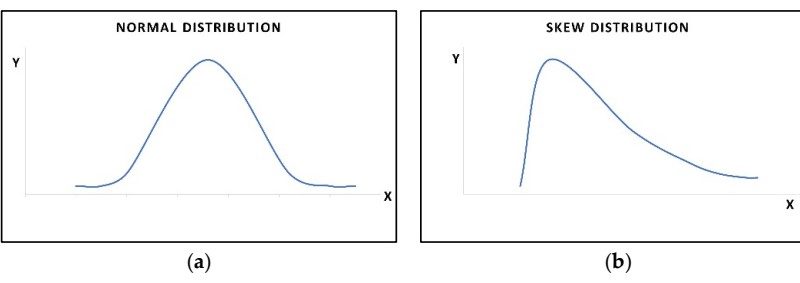

**Figure 8.** (**a**) Normal class distribution, (**b**) skew class distribution.

Similarly, widely used accuracy assessments need to be adopted. Traditionally, the most widespread metric to evaluate the performance of a classification model is accuracy. In the remote sensing community, the kappa coefficient has been considered an advanced evaluation metric in comparison to overall accuracy (Congalton et al., 1983 [58]; Fitzgerald and Lees, 1994 [59]). Nevertheless, Foody [60] explained that the Kappa coefficient is unsuitable for assessing and comparing the accuracy of thematic maps obtained by image classification. This suggests that researchers should abandon the use of the Kappa coefficient in accuracy assessments. In addition, the author encouraged them to use a set of simple evaluation metrics and associated outputs, like estimating accuracy per class and a confusion matrix for evaluation and comparison of the classification accuracy.

As presented in all the above case studies, these metrics are widely adopted but are not reliable for imbalanced data classification. Joshi et al. in 2001 [61] and Weiss in 2004 [62] reported that accuracy is no longer a proper evaluation metric for classification cases with imbalanced data, since the minority class has an insignificant impact on accuracy compared to the majority class. The preliminary results of accuracy presented in this study confirmed the 2009 study of Prati et al. [63], which stated that it is easy to achieve an accuracy of 99.9% in a domain where the majority class has a 99.9% prevalence. All these observations indicate that archaeological ceramics detection is characterized by imbalanced data related to surface ceramics, soil, and crops, where ceramics represent the minority class, and soil and crops represent the majority classes.

Improved classification results would be valuable for further analyses and the development of tools and a workflow to treat imbalanced data or to re-design learning algorithms. At the data level, a possible solution would be rebalancing the class distribution by re-sampling the data space. Meanwhile, at the algorithm level, a solution would be to adapt existing classifier learning algorithms to strengthen learning regarding the small ceramics class. Furthermore, boosting algorithms are considered for future work facing the problem of imbalanced data.

## 6. Conclusions

Our study aimed to investigate whether it is possible to detect archaeological ceramics in an automated way by applying artificial intelligence techniques to high-resolution images captured with UAVs. In addition, we aimed to provide answers regarding the development of a methodology that will perform efficiently in terms of time and accuracy compared to traditional archaeological field surveys. Thus, supervised machine learning algorithms were implemented using RGB and multispectral UAV images.

The overall findings of this study in a simulated environment, utilising the methodology presented by Orengo and Garcia-Molsosa [2], showcased that low-altitude remote sensing sensors can be innovative in archaeological research. The classifiers tend to predict majority classes with high accuracy, while they are useless for predicting minority classes. In our study, a methodology was proposed to overcome this problem and detect surface ceramics using RGB and multispectral drone images.

In this paper, the detection of ceramics was limited to a single cluster of ceramics (one type), as this was the current archaeological record. Nevertheless, the authors expect to investigate the detection of different clusters of ceramics in the same area, i.e., archaeological findings of different chronological periods with different typologies and spectral behaviours. Of course, the detection of various classes of ceramics during the same flight requires a (statistically) significant spectral separability of the different types of ceramics. Controlled and laboratory spectral measurements may provide further insights into this direction (e.g., spectral windows to optimise and enhance the separability of the ceramics).

Future work will include new drone survey campaigns with surface ceramics in the same simulated and known archaeological area. These campaigns will increase the data available for training the algorithms and apply all the methodologies to evaluate and compare the results. Further applications include flights at different heights and further analyses using deep learning algorithms. Other classification improvements will include eliminating random noise, filtering noise, and separability or a combination of all of them, obtaining a combination of new data, modifications of the supervised classifiers that were used, and implementing other boosting algorithms. Evaluating imbalanced ceramics data will also assess the sensitivity of such data using other evaluation measures like F-measure, G-mean, and ROC analysis. These types of measures are ideal evaluation measures because they consider only the positive classes in the performance (True Positive Rate (TPrate) and Positive Predictive Value (PPvalue)). The basic steps of the future research methodology are illustrated in Figure 9.

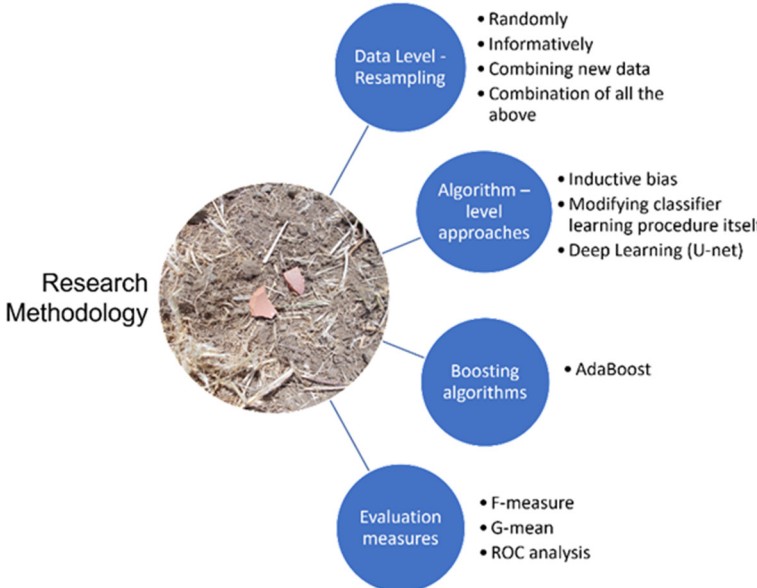

**Figure 9.** Future analysis methodology for treating imbalanced data.



**Author Contributions:** Conceptualization, methodology, A.A. (Argyro Argyrou) and A.A. (Athos Agapiou); writing—original draft preparation, A.A. (Argyro Argyrou); writing—review and editing, A.A. (Athos Agapiou), A.P. and D.D.A.; supervision, A.A. (Athos Agapiou). All authors have read and agreed to the published version of the manuscript.

**Funding:** This research was supported by the ENSURE project (innovative survey techniques for the detection of surface and sub-surface archaeological remains), a Cyprus University of Technology internal funding, as well as the ENGINEER project. ENGINEER received funding from the European Union's Horizon Europe Framework Programme (HORIZON-WIDERA-2021-ACCESS-03, Twinning Call) under the grant agreement No 101079377 and the UKRI under project number 10050486. Disclaimer: The views and opinions expressed are, however, those of the authors only and do not necessarily reflect those of the European Union or the UKRI. Neither the European Union nor the UKRI can be held responsible for them.

**Data Availability Statement:** Not applicable.

**Acknowledgments:** The authors acknowledge the ENSURE project (innovative survey techniques for the detection of surface and sub-surface archaeological remains) and CUT internal funding as well as the ENGINEER project (HORIZON-WIDERA-2021-ACCESS-03, Twinning Call). This paper is part of the PhD dissertation of A. Argyrou, supervised by A. Agapiou, and co-supervised by A.P. and D.D.A. The authors would like to thank the three anonymous reviewers for their criticism and suggestions for improvements during the review process.

**Conflicts of Interest:** The authors declare no conflict of interest. The funders had no role in the design of the study, in the collection, analyses, or interpretation of data; in the writing of the manuscript; or in the decision to publish the results.

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
