# Peer review of "Comparison of Machine Learning Pixel-Based Classifiers for Detecting Archaeological Ceramics"

_drones, doi:10.3390/drones7090578_

Round 1
Reviewer 1 Report
Summary: This article describes an experiment where pottery sherds were placed in a field with good visibility, captured two different types of drone images, and evaluated four different methods for detecting sherds in the data. The results were not particularly successful. The authors identify the reasons for the that, pointing out inadequacies in frequently used measures of accuracy, and suggest future research to address the problem. Drone surveys would be extremely useful if they could identify sherds in the field.
This well-written article clearly explains the procedures, methods, and results (for the most part). The use of an actual physical experiment with pottery sherds is methodologically much better than the more common approach of generating digital data models.
Although they found low accuracy for detecting ceramics with the methods used here, in my opinion, the article definitely merits publication because reports of negative results are very useful. The authors give a convincing theoretical explanation for why the commonly used methods have low accuracy which is intrinsic to the data structure. They also suggest new directions for research.
Some suggestions to improve the manuscript:
Line 24 – could you explain how these results can be innovative. What benefits do they offer?
Lines 75-84 This paragraph uses the terms machine learning (ML) and deep learning (DL) without defining them. Explanations of the terms would be useful to readers who are not intimately familiar with these concepts. Lines 142-149 somewhat cover this same material. Perhaps moving that information earlier in the manuscript would be helpful.
Line 93 – Tell which specific AI techniques are being referred to.
Lines 94-96 What does the phrase “how the environment changed the oldest populations” mean? Do you mean how ancient populations adapted to their environments?
At around Line 152, I was wondering how clustering of ceramics would influence these results or be recognized. Did the experiment include clusters of ceramics, which is how they are commonly found on the surface (especially when used to delineate sites)? The images in Figure 8 suggest so. Perhaps the authors could give their opinion on how clusters of sherds could be detected, and whether the methods would differ from the detection of individual sherds.
Figure 8 needs some additional explanation. Are the future methodologies depicted by the circles on the right to be done in combination or separately, as suggested by the figure? Also, the figure uses abbreviations and terms that are not explained in the text. I would encourage the authors to write a paragraph that explains more about what this figure means. Or perhaps add along caption if that is more appropriate for this journal.
The English is generally quite good. Only minor changes are needed to make it more idiomatic.
Reviewer 2 Report
Please see the attached file.

Maybe minor fine checking. Generally, the english language is good.
Round 2
Reviewer 2 Report
The authors have collaborated towards manuscript improvement